# D-LEAF: Localizing and Correcting Hallucinations in Multimodal LLMs via Layer-to-Head Attention Diagnostics

## Abstract

Multimodal Large Language Models (MLLMs) achieve strong performance on tasks like image captioning and visual question answering, but remain prone to hallucinations, where generated text conflicts with the visual input. Prior work links this partly to insufficient visual attention, but existing attention-based detectors and mitigation typically apply uniform adjustments across layers and heads, obscuring where errors originate. In this paper, we first show these methods fail to accurately localize problematic layers. Then, we introduce two diagnostics: Layer Image Attention Entropy (LIAE) which flags anomalous layers, and Image Attention Focus (IAF) which scores attention heads within those layers. Analysis shows that LIAE pinpoints faulty layers and IAF reliably ranks heads that warrant correction. Guided by these signals, we propose Dynamic Layer-wise Entropy and Attention Fusion (D-LEAF), a task-agnostic, attention-guided method that dynamically localizes and corrects errors during inference with negligible overhead. Results show our D-LEAF delivers a 53% relative improvement on standard captioning benchmarks, and on VQA both accuracy and F1-score improve by approximately 4%, substantially suppressing hallucinations while preserving efficiency.

## 1 Introduction

Multimodal Large Language Models (MLLMs) have gained increasing attention for their ability to process and integrate visual and textual information. This design allows them to achieve strong performance on a variety of vision-language tasks, such as image captioning, visual question answering, and text-to-image generation (Chen et al., 2023; Zhu et al., 2023; Liu et al., 2024a). However, MLLMs often produce content that contradicts the image or the instructions, which is known as hallucination. These inconsistencies often lead to reliability issues in practical applications, particularly in domains where accuracy and factual consistency are critical (He et al., 2023; Guo et al., 2024; Zhou et al., 2025).

Traditional strategies to mitigate hallucinations in vision-language models involve instruction fine-tuning or reinforcement learning on carefully curated datasets (Gunjal et al., 2024; Jiang et al., 2024). Although effective, these approaches are typically resource-intensive and difficult to scale. To overcome these challenges, recent research has shifted to inference-time methods, mitigating hallucinations by enhancing semantic stability (Chen et al., 2025; Wang et al., 2025a; Tang et al., 2025) or applying contrastive decoding techniques (Wang et al., 2024; Liang et al., 2025; Jiang et al., 2025a) to adjust the distribution of the final output logits. Although these methods are more effective than training-based algorithms, they still cannot sufficiently eliminate hallucinations and incur a higher inference latency relative to the baseline.

In addition, a deeper limitation is mechanistic: prior methods rarely identify *where* hallucinations arise in the attention stack. Several studies implicate over-reliance on the language stream (e.g., "anchor patterns" (Huang et al., 2024) or "textual inertia" (Liu et al., 2024b)) and respond with global adjustments that increase visual weighting (Sarkar et al., 2025; Jiang et al., 2025b). However, in practice, we find that these interventions frequently apply undifferentiated suppression across all

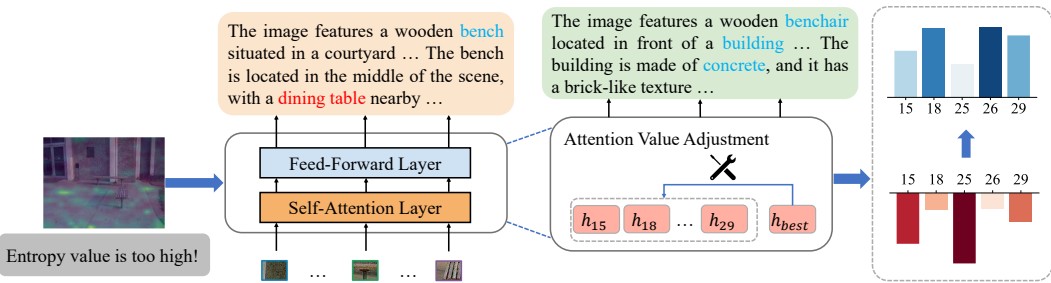

Figure 1: The workflow of D-LEAF. During inference, when a layer's attention-module entropy exceeds a dynamic threshold, D-LEAF then corrects the attention heads exhibiting insufficient visual focus, suppressing hallucinations (e.g., the phrase "dining table").

selected attention modules. This can disrupt correctly functioning heads and thus limit hallucination reduction (see in Figure 2).

To address these issues, we adopt a *localize before correct* strategy. We first introduce two complementary diagnostics that operate during the forward pass: (1) Layer Image-Attention Entropy (LIAE), which flags unreliable layers; and (2) Image-Attention Focus (IAF), which identifies the specific attention heads within those anomalous layers that require correction. Guided by LIAE and IAF, we propose D-LEAF (Dynamic Layer-wise Entropy and Attention Fusion), a lightweight and plug-and-play method that dynamically identifies unreliable attention components and applies selective, fused corrections to only the flagged heads, avoiding blanket suppression. As illustrated in Figure 1, when the attention entropy of a layer exceeds a dynamic threshold, D-LEAF pinpoints low focus heads and injects fused corrective signals, which suppress hallucinated content (e.g., removing the spurious phrase 'dining table') while preserving faithful details.

In experiments, we evaluate D-LEAF on three representative MLLM architectures across three standard multimodal hallucination benchmarks, comparing against six state-of-the-art correction methods. Results show D-LEAF consistently delivers the strongest suppression, reducing hallucinations by up to 53% versus the baseline and improving VQA accuracy and F1 by approximately 4%, with only 8% throughput drop relative to greedy decoding. These results demonstrate that D-LEAF strikes an optimal balance between factual reliability, descriptive detail, and high inference speed. Our contributions are summarized as follows:

- First, we analyze prior attention-head-based suppression methods and show that (i) some attention heads can focus on the correct image information and (ii) blindly suppressing all heads across layers can harm correct ones, leading to ineffective hallucination mitigation. To address this, we propose two inference-time diagnostics LIAE and IAF to dynamically and precisely localize anomalous layers and specific heads requiring correction.

- Second, we propose a novel method, called D-LEAF. D-LEAF is a lightweight plug-and-play method that suppresses hallucinations through layer-by-layer corrections during inference: LIAE flags problematic layers, and IAF selects heads to receive fused corrective signals.

- Third, we validate the effectiveness of D-LEAF through extensive experiments on three leading MLLMs in three multimodal hallucination benchmarks, achieving up to a 53% reduction in hallucinations with only 8% throughput overhead relative to greedy decoding, without relying on additional tools.

## 2 RELATED WORK

**Hallucination and Mitigation in MLLMs.** In natural language processing, hallucinations originally denote generated content that is inconsistent with the context or facts (Huang et al., 2025). In MLLMs, this manifests as factual errors, incorrect image descriptions, or misidentified object attributes/relationships (Liu et al., 2024a). Previous research on hallucination mitigation can be divided mainly into two categories: training-based algorithms and training-free algorithms. Training-based methods apply visual instruction tuning (Gunjal et al., 2024), external expert guidance (Chen et al., 2024), or reinforcement learning from human feedback (RLHF) (Sun et al., 2023), but they

typically require substantial compute and are difficult to deploy in resource-constrained settings. Thus, lightweight training-free methods have attracted growing interest. A prominent line is contrastive decoding, which mitigates spurious output by comparing model predictions under varying conditions. For example, VCD (Leng et al., 2024) contrasts the output distributions conditioned on original and distorted visual inputs to identify and suppress hallucinated content; MoLE (Liang et al., 2025) employs a Mixture of Experts for inter-layer contrast decoding; DAMRO (Gong et al., 2024) reduces the impact of background outlier tokens; OPERA (Huang et al., 2024) performs multiple rollbacks combined with token aggregation to suppress hallucinations; DoLA (Chuang et al., 2023) leverages layer-wise contrasts to enhance factuality; and HALC (Chen et al., 2024) contrasts output distributions across different visual contexts and uses visual matching scores to guide beam-search candidate selection. Despite their effectiveness, these approaches still introduce additional decoding overhead, e.g., HALC incurs a 2.4× increase in inference time compared to standard greedy decoding. Thus, this motivates us to design a lightweight, plug-and-play method without relying on additional tools.

**Interpretability-driven Mitigation in MLLMs Hallucination.** Numerous studies have examined the underlying causes of hallucinations to guide the development of more fine-grained architecture-level suppression methods. Reported factors include excessive prior knowledge of LLM (Liu et al., 2024b), insufficient attention to images (Jiang et al., 2025b; Sarkar et al., 2025; You et al., 2025), and excessive attention to summary words (Huang et al., 2024). Among these mitigation strategies, attention-head-based hallucination suppression methods show promise. For example, ASCD (Wang et al., 2025b) employs positive and negative steering as two complementary mechanisms to adapt the internal attention distributions of the model. AD-HH (Yang et al., 2025) first identifies the heads prone to hallucination offline and then detects and suppresses these heads in real time during the model's forward pass. In contrast, SPIN (Sarkar et al., 2025) and SVAR (Jiang et al., 2025b) indiscriminately mute a subset of heads in specific layers to force the model to focus more on visual input. However, because these approaches apply uniform corrections across all layers, they lack flexibility and can still fail to eliminate hallucinations in certain cases, as illustrated in Figure 2. To address these issues, we conducted a systematic analysis of attention-module behavior during the forward pass of the model and introduced LIAE for layer-wise detection of problematic heads within each decoder module. By applying targeted per-layer corrections to these specific attention heads, our method more precisely suppresses hallucinations.

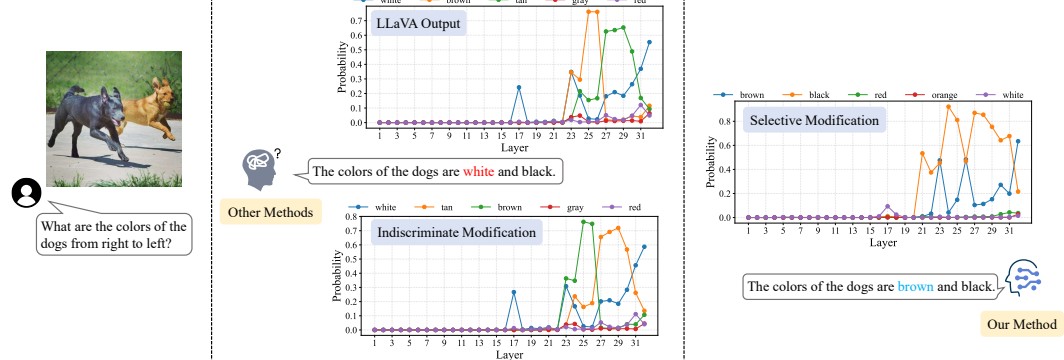

Figure 2: A motivating example of using selective attention correction in a visually ambiguous scenario.

## 3 UNDERSTANDING MLLM HALLUCINATION

In this section, we present empirical analyses to investigate the internal mechanisms behind hallucination in MLLMs from the perspective of attention. Although prior attention-based hallucination mitigation methods have achieved promising results, we observe that they still fail to produce correct answers in semantically ambiguous scenarios, as illustrated in Figure 2. We therefore re-examine these approaches in detail. We hypothesize that *indiscriminately suppression of attention heads across all layers reduces hallucinations but biases the model toward generating shorter outputs.* We aim to answer these three research questions: (i) Are poorly performing heads uniformly dis-

tributed across layers?; (ii) How does head suppression reduce hallucinations?; (iii) What costs does suppression of poorly performing heads incur?

## 3.1 INDISCRIMINATE CORRECTION LEADS TO ERRORS

**Are poorly performing heads uniformly distributed across layers?** Prior attention-correction strategies such as SPIN (Sarkar et al., 2025) and SVAR (Liu et al., 2024b) suppress the lowest-scoring heads at the intra-layer level. We argue that ignoring inter-layer head performance can inadvertently suppress functionally correct heads, and thus fail to eliminate hallucinations. As illustrated in the middle of Figure 2, applying SPIN to LLaVA in a visually ambiguous scenario does not prevent the model's hallucinated outputs. We hypothesize that this phenomenon arises because poorly performing attention heads are not uniformly distributed across all layers, but instead cluster within specific layers.

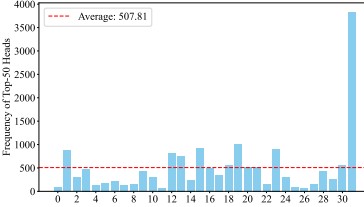

Figure 3: Distribution of abnormal attention heads across layers.

To further verify this hypothesis, we randomly sampled 500 images from the COCO2014 validation set and extracted the hallucinated tokens generated by the model. We rank every attention head across all layers by their SPIN scores, select the 50 lowest performing heads, and visualize their distribution to reveal how these underperformers cluster across layers. As illustrated in Figure 3, this distribution markedly deviates from uniformity, with the majority of layers falling below the mean. This suggests that indiscriminately modifying the lowest-scoring anomalous attention heads across all layers is not a principled or effective strategy.

Based on the above assumptions, in the context of Figure 2, we rank all attention heads globally by their SPIN scores, select the $k$ worst performing heads (using the same $k$ as SPIN) and suppress them. As shown on the right side of Figure 2, this global suppression successfully prevents the model from hallucination. These results motivate us to design a metric to dynamically localize anomalous attention heads during the model's forward pass.

## 3.2 MUTING LOW-FOCUS HEADS REDUCES HALLUCINATIONS AT THE COST OF ACCURACY

In this part, we answer the questions 'How does head suppression reduce hallucinations?' and 'What costs does suppression of poorly performing heads incur?'. Inspired by (Li et al., 2023a), we investigate whether there exist attention heads that can correctly capture image content. Following (Rohrbach et al., 2018), we adopt the CHAIR metrics, namely $CHAIR_S$ and $CHAIR_I$, as defined in equation 6, and conduct experiments on LLaVA-7B. The detailed definitions of these metrics are provided in the Appendix B.1.

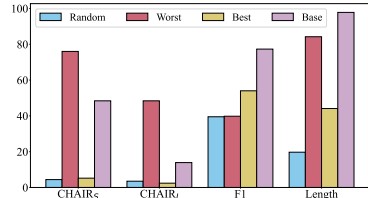

Figure 4: The impact of different suppression methods in LLaVA.

We first partition the attention heads by ranking them within each layer according to their cumulative attention over the image tokens. Based on this ranking, we suppress 15% of the heads per layer, yielding four experimental settings: (i) no intervention, (ii) suppressing the heads with the highest attention scores, (iii) suppressing the heads with the lowest attention scores, and (iv) randomly suppressing heads. We then evaluate each setting using four metrics: $CHAIR_S$, $CHAIR_I$, F1, and output length (Length), as illustrated in Figure 4.

Our results demonstrate that attention heads with high focus on image tokens indeed contribute to visual understanding, as suppressing them leads to a substantial increase in hallucination rates, approximately twice that of the baseline. Conversely, suppressing heads with low image attention can significantly reduce hallucination rates, but this comes at the cost of shorter outputs and a decrease in F1 scores. We attribute this drawback to the indiscriminate suppression of heads across all layers, as discussed above. Moreover, our experiments reveal that modifying only a small subset of attention heads can substantially alter the model's output, which is consistent with the findings in Kang et al. (2025a;b).

Building on these observations, we hypothesize that increasing the visual focus of underperforming attention heads can reduce hallucinations, and that self-correction can be achieved by directly leveraging higher-scoring heads within the abnormal layers.

# 4 D-LEAF

We investigate in depth the internal mechanisms behind hallucinations in the last section. In this section, we introduce the Dynamic Layer-wise Entropy and Attention Fusion (D-LEAF) framework to mitigate hallucinations. Our method dynamically detects anomalous behaviors in the MHA modules of MLLMs during the forward pass and applies real-time corrections to the identified problematic components, thereby improving the reliability of the model's outputs, as shown in Figure 23. We begin by introducing a novel metric Layer Image Attention Entropy (LIAE) for detecting anomalous behavior within each decoder module and describe how, once a module is flagged, we use a additional indicator Image Attention Focus (IAF) to pinpoint the exact attention heads that need to be corrected. We also present significance tests and correlation analyses for these metrics (see Appendix C for introductions to these tools). Finally, we detail the complete algorithmic workflow. We have verified the validity and effectiveness of these indicators in Appendix D.2 and replicated all analyses presented in this section on Shikra to validate the generalizability of our proposed metrics; details are provided in Appendix D.3.

## 4.1 DYNAMIC LAYER SELECTION

As discussed before, prior methods typically rank attention heads within each layer and directly suppress those with the lowest scores. However, this intra-layer ranking ignores cross-layer context: if a given layer already exhibits higher overall attention scores than other layers, its comparatively weaker heads may still be performing adequately. As a result, suppressing them indiscriminately can fail to reduce and may even exacerbate hallucinations.

Although several layer selection methods have been proposed, particularly in contrastive decoding, for instance, DeCo (Wang et al., 2024), which mixes the maximum-probability logits from a selected layer with those of the final layer, and MoLE (Liang et al., 2025), which adopts a hybrid selection strategy by comparing intermediate logits against the final layer, these approaches rely on a baseline, typically the output of the last layer. However, such a baseline is infeasible for attention-head-based corrections, since modifications to attention occur during the forward pass and each layer's perturbation directly propagates to subsequent computations. This motivates the need for a new dynamic metric to detect abnormal layers.

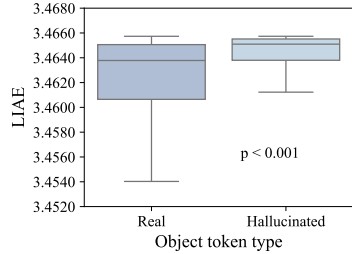

Figure 5: LIAE Distribution across object token types in MiniGPT-4.

Prior work has shown that, during inference in MLLMs, only a subset of attention heads are functionally engaged, and that elevated attention-head entropy is associated with a higher likelihood of hallucination (Kang et al., 2025b; Jiang et al., 2025b). Therefore, to capture the overall state of all attention heads in the current layer, we first introduce the Maximum Attention Matrix (MAM). For the $i$-th token of an MLLM output $t_i$, we define the MAM of the $l$-th layer:

$$\text{MAM}_n^{(l)} = \max_{h=1,\ldots,H} \text{A}_{h,n}^{(l)}, \quad n = 1,\ldots,N. \tag{1}$$

where $A_{h,n}^{(l)}$ represents the attention of the $h$-th attention head in the $l$-th layer to the image token $n$. Each entry of the MAM at layer $l$ represents, for a given image token $n$, the highest attention score that any of that layer's attention heads assigns to $n$.

With MAM, we introduce a metric called Layer Image Attention Entropy (LIAE), which quantifies whether a given layer contains attention heads exhibiting overly diffuse focus and therefore require correction.

$$\text{LIAE}^{(l)} = -\sum_{n=1}^{N} \text{P}(\text{MAM}_n^{(l)})\log\text{P}(\text{MAM}_n^{(l)}) \tag{2}$$

To validate the effectiveness of the metrics we proposed, we first randomly selected a subset of 500 images from the COCO 2014 validation set (Lin et al., 2014). We chose MiniGPT-4 for subsequent analysis. To show whether LIAE can significantly distinguish the differences between layers when the model generates real words and hallucinated words, we use greedy search in the decoding process of the above model to generate captions for the selected images, prompted by "Please help me describe the image in detail." We use the ground truth annotation to identify the real and halluci-nated words. We then calculated and plotted the distribution of LIAE when the model generated hallucinated words and real words.

To evaluate the significance of these metrics, which independent and non-normally distributed across real and hallucinated tokens, we apply the Wilcoxon signed-rank test (Wilcoxon, 1992). With $p < 0.001$, in Figure 5, we confidently observe in these two models that hallucinated tokens exhibit significantly higher LIAE compared to real tokens.

Accordingly, based on the above experiments, we use LIAE to localize layers that contain anomalous attention heads. For completeness, we report in Appendix D.2 additional experiments that use Layer Image Attention Focus (LIAF) and a hybrid of the two (LIAS) as alternative criteria for abnormal-layer detection, together with ablations. We find that LIAE is more sensitive and achieves the best performance; therefore, we adopt LIAE as the sole metric for abnormal-layer localization.

## 4.2 ATTENTION HEAD LOCALIZATION

After pinpointing abnormal layers, we must identify the specific attention heads within them that require modification. Motivated by the "text inertia" phenomenon (Liu et al., 2024b), we introduce Image Attention Focus (IAF), a metric that quantifies the extent to which each attention head attends to image tokens (i.e., visual regions).

$$\text{IAF}_h^{(l)} = \sum_{n=1}^{N} \mathbf{A}_{h,n}^{(l)} \tag{3}$$

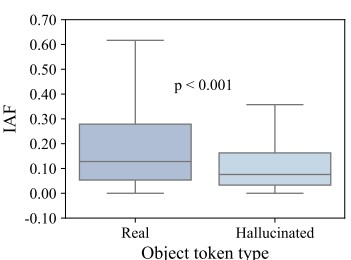

Figure 6: IAF Distribution across object token types in MiniGPT-4.

Similarly, to validate the effectiveness of these two metrics, we repeated the Wilcoxon signed-rank test described above in MiniGPT-4. As shown in Figure 6, we plotted the distributions and found that, at p < 0.001, every attention head exhibits sig-nificantly higher IAF for real tokens than for hallucinated ones.

Analogous to LIAE, to validate the soundness of our metric, we also evaluate Image Attention Entropy (IAE) and a hy-brid of IAF and IAE for identifying and correcting anomalous attention heads; implementation details are provided in Ap-pendix D.2. Across multiple model architectures, however, we find that using IAF alone consistently achieves the strongest hallucination suppression.

We also verified the reliability of our metrics on other model architectures, with specific results provided in Appendix D.3.

## 4.3 MIXED ATTENTION MATRIX CORRECTION

Building on the aforementioned layer selection and head localization, we could detect anomalous heads during the model's forward pass. Our prior experiments indicate that disrupting attention heads with stronger focus on visual tokens causes disproportionately greater degradation and, in particular, increases the model's propensity to produce hallucinated outputs. Moreover, directly suppressing attention heads with insufficient attention leads to a decline in the quality of the model's output content. Therefore, we hypothesis that hallucinations stem in part from certain heads al-locating insufficient attention to visual regions. To remedy this, we introduce a mixed correction approach that leverages well-performing heads to refine the under-performing heads.

$$A_{h,v}^l = \gamma A_{best,v}^l + (1 - \gamma) A_{h,v}^l \tag{4}$$

where $A_{h,v}^l$ and $A_{best,v}^l$ denote, respectively, the visual-token submatrices of the attention matrices for the heads requiring correction and the highest-scoring heads at layer $l$.

## 4.4 DYNAMIC LAYER-WISE ENTROPY AND ATTENTION FUSION

Based on these findings, we propose a two-stage detection and correction algorithm to effectively suppress model hallucinations. Unlike traditional attention-head-based hallucination suppression methods, we introduce a dynamic detection baseline Best Attention Score (BAS) that monitors the model's forward pass and pinpoints the specific layer where anomalies arise.

$$\text{BAS} = \min(\text{BAS}, \text{LIAE}^{(l)}), \quad l = 1, 2, \ldots, L. \tag{5}$$

BAS denotes the best score of the layer encountered so far in the forward pass, and it is updated incrementally. During inference, if the current layer's LIAE is lower than BAS, we set BAS equal to LIAE; if LIAE exceeds BAS, we deem the attention heads in this layer to be underperforming and therefore in need of correction. We then rank all heads in the current layer by their IAF values, select the $n$ worst-performing heads, and update their attention matrices according to equation 4.

Our dynamic localization-and-correction algorithm not only suppresses hallucinations at a finer granularity within the model's architectural layers but also leverages inter-layer relationships to pinpoint anomalies with greater accuracy. The detailed workflow of the algorithm and its pseudo-code is provided in the Appendix D.5.

## 5 EXPERIMENT

### 5.1 EXPERIMENTAL SETTINGS

**Datasets.** We conducted a detailed evaluation of our proposed algorithm using three established benchmarks: CHAIR Rohrbach et al. (2018), POPE Li et al. (2023b), and MMHal-Bench Sun et al. (2023). These benchmarks were used to assess the effectiveness of our method in suppressing model hallucinations. Detailed descriptions of the benchmarks are provided in Appendix B.1.

**Models and Baselines.** We evaluate our method D-LEAF on 5 models: LLaVA-1.5 (7B) (Liu et al., 2024a), MiniGPT-4 (Zhu et al., 2023), Shikra (Chen et al., 2023), InstructBLIP (Dai et al., 2023) and Qwen-VL Bai et al. (2023). And we compare it with several existing SOTA training-free hallucination suppression algorithms, including Greedy Search and Nuclear Sampling, SPIN (Sarkar et al., 2025), PAI (Liu et al., 2024b), VCD (Leng et al., 2024), and DAMRO (Gong et al., 2024). More details are provided in Appendix B.2.

**Evaluation Metric.** CHAIR provided two indicators: $C_I$, which indicates the hallucination rate at the instance-level, and $C_S$, which the hallucination rate at the sentence-level. They are calculated with the following equation:

$$\text{CHAIR}_I = \frac{|\{\text{hallucinated objects}\}|}{\text{all mentioned objects}}, \quad \text{CHAIR}_S = \frac{|\{\text{captions with hallucinated objects}\}|}{\text{all captions}}. \tag{6}$$

**Implementation Details.** All experiments are run in Pytorch using vGPU-48GB. We used a batch size of 1 and set the model's maximum output length to 512 tokens. For each configuration, we report the average and standard deviation over 3 runs with different random seeds $\{42, 927, 111\}$ in POPE and 5 runs with different random seeds $\{42, 3, 11, 927, 111\}$ in CHAIR. For the image captioning and VQA tasks, we set the number of attention heads that need to be modified on each layer between 3 and 5, $\gamma$ is set within the range of 0.7 to 0.9.

### 5.2 MAIN RESULTS

**Long Sequence Hallucination Evaluation.** We evaluate the CHAIR result of five models, as presented in Table 1. Our D-LEAF method significantly outperforms previous state-of-the-art approaches across all metrics on hallucination and models. Specifically, on MiniGPT-4 our model

Table 1: CHAIR hallucination evaluation results. The best result is highlighted in bold, and the second-best is underlined. The values reported are the mean performance.

| Method | LLaVA | | MiniGPT-4 | | Shikra | | InstructBLIP | | Qwen-VL | |
|---|---|---|---|---|---|---|---|---|---|---|
| | $C_S$ | $C_I$ | $C_S$ | $C_I$ | $C_S$ | $C_I$ | $C_S$ | $C_I$ | $C_S$ | $C_I$ |
| Greedy | 47.08 $_{\pm 1.54}$ | 13.00 $_{\pm 0.59}$ | 34.00 $_{\pm 1.98}$ | 10.82 $_{\pm 0.59}$ | 54.64 $_{\pm 2.84}$ | 14.96 $_{\pm 1.37}$ | 48.12 $_{\pm 2.98}$ | 14.18 $_{\pm 1.12}$ | 46.88 $_{\pm 1.56}$ | 12.72 $_{\pm 0.36}$ |
| Sampling | 53.44 $_{\pm 2.21}$ | 16.30 $_{\pm 1.26}$ | 33.80 $_{\pm 1.17}$ | 11.78 $_{\pm 0.62}$ | 57.10 $_{\pm 1.80}$ | 16.14 $_{\pm 0.88}$ | 47.04 $_{\pm 1.47}$ | 13.24 $_{\pm 0.77}$ | 47.04 $_{\pm 1.47}$ | 13.24 $_{\pm 0.77}$ |
| VCD | 55.38 $_{\pm 1.17}$ | 15.20 $_{\pm 1.21}$ | – | – | 55.16 $_{\pm 2.25}$ | 14.96 $_{\pm 1.26}$ | – | – | 51.40 $_{\pm 2.21}$ | 13.62 $_{\pm 0.75}$ |
| PAI | 35.28 $_{\pm 1.69}$ | 9.46 $_{\pm 0.67}$ | 27.92 $_{\pm 1.47}$ | 10.06 $_{\pm 0.89}$ | 54.64 $_{\pm 2.51}$ | 14.02 $_{\pm 1.49}$ | 59.24 $_{\pm 2.00}$ | 16.10 $_{\pm 0.67}$ | 47.64 $_{\pm 2.26}$ | 12.92 $_{\pm 0.68}$ |
| DAMRO | 46.44 $_{\pm 1.66}$ | 12.78 $_{\pm 0.60}$ | – | – | – | – | – | – | – | – |
| SPIN | 29.04 $_{\pm 2.46}$ | 8.70 $_{\pm 0.55}$ | 24.56 $_{\pm 1.62}$ | 9.40 $_{\pm 1.74}$ | 38.56 $_{\pm 2.35}$ | 10.88 $_{\pm 0.56}$ | 48.80 $_{\pm 2.65}$ | 14.04 $_{\pm 1.23}$ | 33.72 $_{\pm 2.54}$ | 9.42 $_{\pm 0.48}$ |
| D-LEAF | **23.44** $_{\pm 2.63}$ | **6.72** $_{\pm 0.49}$ | **11.56** $_{\pm 1.69}$ | **4.72** $_{\pm 0.95}$ | **26.35** $_{\pm 1.32}$ | **10.62** $_{\pm 0.87}$ | **22.44** $_{\pm 2.75}$ | **8.48** $_{\pm 5.93}$ | **25.24** $_{\pm 1.55}$ | **7.96** $_{\pm 1.65}$ |

Table 2: Quantitative comparison on Muti-turn POPE. The best result is highlighted in bold, and the second-best is underlined. The values reported are the mean performance.

| Model | Method | Random | | Popular | | Adversarial | |
|---|---|---|---|---|---|---|---|
| | | Accuracy | F1 | Accuracy | F1 | Accuracy | F1 |
| LLaVA | Greedy | 86.63 $_{\pm 0.78}$ | 85.32 $_{\pm 0.92}$ | 79.22 $_{\pm 0.23}$ | 78.23 $_{\pm 0.66}$ | 76.98 $_{\pm 0.16}$ | 76.71 $_{\pm 0.41}$ |
| | Sampling | 83.94 $_{\pm 0.59}$ | 83.30 $_{\pm 0.44}$ | 76.30 $_{\pm 0.15}$ | 75.05 $_{\pm 0.35}$ | 72.70 $_{\pm 1.61}$ | 73.44 $_{\pm 2.85}$ |
| | PAI | 77.02 $_{\pm 4.62}$ | 72.96 $_{\pm 6.94}$ | 75.68 $_{\pm 0.12}$ | 72.65 $_{\pm 0.25}$ | 75.07 $_{\pm 0.55}$ | 72.41 $_{\pm 0.75}$ |
| | DAMRO | 86.67 $_{\pm 0.87}$ | 85.53 $_{\pm 1.09}$ | 79.23 $_{\pm 0.28}$ | 78.28 $_{\pm 0.71}$ | 77.01 $_{\pm 0.17}$ | 76.74 $_{\pm 0.42}$ |
| | SPIN | 86.29 $_{\pm 0.16}$ | 84.80 $_{\pm 0.38}$ | 81.81 $_{\pm 2.85}$ | 80.47 $_{\pm 2.60}$ | 80.31 $_{\pm 4.33}$ | 79.47 $_{\pm 4.01}$ |
| | D-LEAF | **87.76** $_{\pm 0.47}$ | **86.65** $_{\pm 0.72}$ | **84.75** $_{\pm 3.21}$ | **83.67** $_{\pm 3.00}$ | **84.94** $_{\pm 3.35}$ | **82.09** $_{\pm 2.87}$ |
| MiniGPT-4 | Greedy | 70.17 $_{\pm 3.68}$ | 68.33 $_{\pm 3.82}$ | 64.01 $_{\pm 4.39}$ | 63.16 $_{\pm 5.57}$ | 63.06 $_{\pm 2.48}$ | 62.52 $_{\pm 1.17}$ |
| | Sampling | 70.34 $_{\pm 5.48}$ | 60.96 $_{\pm 9.00}$ | 62.92 $_{\pm 5.22}$ | 54.41 $_{\pm 8.31}$ | 60.30 $_{\pm 4.70}$ | 55.52 $_{\pm 7.78}$ |
| | PAI | 68.45 $_{\pm 9.67}$ | 64.12 $_{\pm 7.14}$ | 59.66 $_{\pm 8.49}$ | 57.31 $_{\pm 6.45}$ | 60.93 $_{\pm 6.52}$ | 56.38 $_{\pm 5.42}$ |
| | SPIN | 72.89 $_{\pm 2.44}$ | 69.04 $_{\pm 2.19}$ | 66.10 $_{\pm 2.36}$ | 64.33 $_{\pm 4.18}$ | 64.88 $_{\pm 3.05}$ | **67.34** $_{\pm 2.66}$ |
| | D-LEAF | **74.95** $_{\pm 0.62}$ | **72.17** $_{\pm 0.77}$ | **67.51** $_{\pm 0.09}$ | **66.02** $_{\pm 0.15}$ | **67.62** $_{\pm 0.41}$ | 65.17 $_{\pm 0.46}$ |
| Shikra | Greedy | 80.75 $_{\pm 0.56}$ | 80.35 $_{\pm 0.54}$ | 74.95 $_{\pm 1.88}$ | 75.66 $_{\pm 1.22}$ | 73.37 $_{\pm 1.97}$ | 75.89 $_{\pm 1.03}$ |
| | Sampling | 81.67 $_{\pm 0.65}$ | 81.55 $_{\pm 0.69}$ | 77.58 $_{\pm 0.36}$ | 78.67 $_{\pm 0.51}$ | 72.94 $_{\pm 1.45}$ | 75.37 $_{\pm 0.68}$ |
| | PAI | 71.00 $_{\pm 0.25}$ | 74.03 $_{\pm 0.22}$ | 70.10 $_{\pm 0.39}$ | 73.38 $_{\pm 0.60}$ | 64.97 $_{\pm 0.62}$ | 70.65 $_{\pm 0.09}$ |
| | SPIN | 64.68 $_{\pm 0.03}$ | 61.66 $_{\pm 0.78}$ | 59.73 $_{\pm 0.16}$ | 60.06 $_{\pm 0.61}$ | 58.17 $_{\pm 0.11}$ | 61.09 $_{\pm 0.71}$ |
| | D-LEAF | **82.36** $_{\pm 1.12}$ | **83.32** $_{\pm 1.14}$ | **79.12** $_{\pm 0.61}$ | **79.89** $_{\pm 0.73}$ | **75.15** $_{\pm 2.10}$ | **76.37** $_{\pm 1.45}$ |
| InstructBLIP | Greedy | 86.16 $_{\pm 0.65}$ | 84.54 $_{\pm 0.86}$ | 84.48 $_{\pm 1.10}$ | 83.03 $_{\pm 1.28}$ | 81.95 $_{\pm 0.25}$ | 80.75 $_{\pm 0.46}$ |
| | Sampling | 79.36 $_{\pm 0.20}$ | 78.36 $_{\pm 0.31}$ | 76.55 $_{\pm 0.32}$ | 76.17 $_{\pm 0.18}$ | 74.74 $_{\pm 0.20}$ | 74.93 $_{\pm 0.05}$ |
| | PAI | 86.34 $_{\pm 0.44}$ | 84.66 $_{\pm 0.63}$ | 84.80 $_{\pm 0.94}$ | 83.22 $_{\pm 1.08}$ | **82.69** $_{\pm 0.03}$ | **81.32** $_{\pm 0.12}$ |
| | SPIN | 86.51 $_{\pm 0.55}$ | 85.06 $_{\pm 0.76}$ | 85.21 $_{\pm 1.07}$ | 83.91 $_{\pm 1.24}$ | 81.97 $_{\pm 0.19}$ | 80.96 $_{\pm 0.42}$ |
| | D-LEAF | **86.67** $_{\pm 0.57}$ | **85.23** $_{\pm 0.74}$ | **85.32** $_{\pm 0.96}$ | **84.06** $_{\pm 1.05}$ | 82.10 $_{\pm 0.24}$ | 81.09 $_{\pm 0.42}$ |
| Qwen-VL | Greedy | 89.56 $_{\pm 0.24}$ | 89.20 $_{\pm 0.28}$ | 86.93 $_{\pm 0.05}$ | 86.76 $_{\pm 0.02}$ | 82.81 $_{\pm 0.11}$ | 83.25 $_{\pm 0.02}$ |
| | Sampling | 85.41 $_{\pm 0.32}$ | 84.82 $_{\pm 0.38}$ | 81.49 $_{\pm 0.25}$ | 81.42 $_{\pm 0.14}$ | 77.82 $_{\pm 1.23}$ | 72.24 $_{\pm 0.62}$ |
| | PAI | 89.34 $_{\pm 0.11}$ | 88.89 $_{\pm 0.11}$ | 86.94 $_{\pm 0.30}$ | 86.80 $_{\pm 0.22}$ | 82.88 $_{\pm 0.57}$ | 83.32 $_{\pm 0.40}$ |
| | SPIN | 88.49 $_{\pm 0.03}$ | 88.22 $_{\pm 0.06}$ | 84.66 $_{\pm 1.13}$ | 84.86 $_{\pm 0.99}$ | 80.49 $_{\pm 0.80}$ | 81.48 $_{\pm 0.62}$ |
| | D-LEAF | **89.59** $_{\pm 0.27}$ | **89.21** $_{\pm 0.22}$ | **87.42** $_{\pm 0.13}$ | **87.18** $_{\pm 0.11}$ | **83.19** $_{\pm 0.30}$ | **83.50** $_{\pm 0.19}$ |

achieves a 53% reduction in $C_S$ and a 57% reduction in $C_I$ compared to SPIN, highlighting the effectiveness of D-LEAF in mitigating object hallucinations in long text generation tasks.

**Multi-turn Hallucination Evaluation.** We use a multi-turn POPE evaluation to increase the difficulty of this task, the result is shown in Table 2. D-LEAF consistently performs best across each part of the POPE across the five models. Notably, on LLaVA, MiniGPT-4 and Shikra, our model outperforms the baseline by approximately 5%, and on the other two models, D-LEAF also surpasses the current state-of-the-art by 1%. The results indicated that our D-LEAF could achieve good results in long context VQA tasks.

**GPT-4 Assisted Hallucination Evaluation in Comprehensive General Scenarios.** We use MMHal-Bench and GPT-4 assist to evaluate the performance of D-LEAF in more complex scenarios. From Figure 7, the experimental results indicate that our method could achieve better results in all three models, especially in LLaVA 7B. While for more image-based question types, like attributes and adversarial objects, our method did not achieve a noticeable improvement in MiniGPT-4

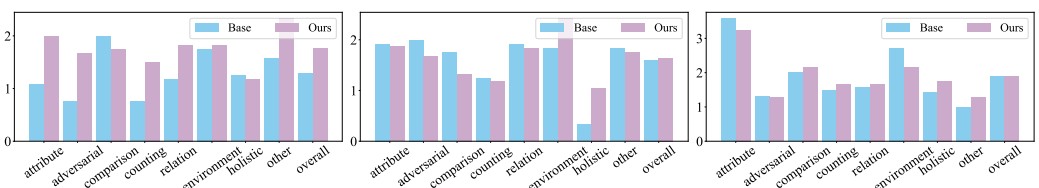

Figure 7: MMHal-Bench Evaluation on LLaVA, MiniGPT-4 and Shikra.

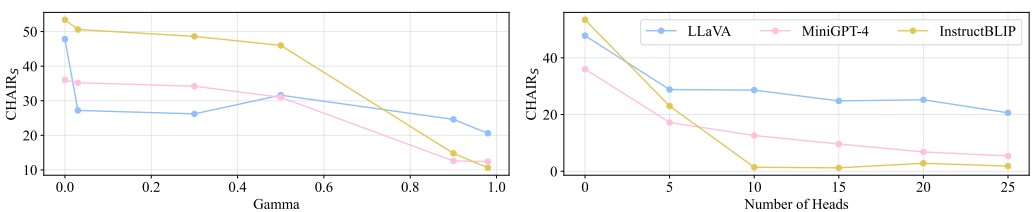

Figure 9: Ablation Study results for hyperparamter $\gamma$ and $n$.

and Shikra. In the average performance across the eight evaluation dimensions, there is a certain degree of improvement compared to the baseline after incorporating D-LEAF.

**Throughput Estimation.** To evaluate whether our algorithm maintains real-time efficiency without incurring significant throughput loss, we measured the token-per-second generation rate of LLaVA under different algorithms, as shown in Figure 8. Our method showed the least reduction in throughput compared to the baseline. It outperformed other state-of-the-art techniques, including attention-head correction methods such as SPIN and PAI. We repeated this experiment on MiniGPT-4 and Shikra and observed consistent results, as detailed in Appendix B.3.

In Appendix E, we provide visualizations across diverse MLLMs to further present instances of hallucination corrections by our method.

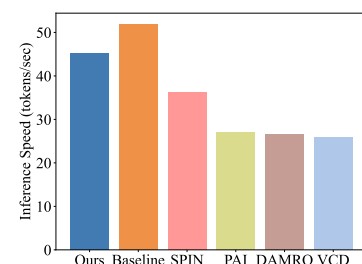

Figure 8: Throughput comparison with existing methods in LLaVA.

### 5.3 ABLATION STUDY

D-LEAF incorporates two primary hyperparameters: $\gamma$ and the number of heads $n$. We present the effect of varying two parameters on CHAIR$_S$ in the main text, as illustrated in Figure 9. The results demonstrate the strong robustness of our method: across a wide range of $\gamma$ (0.03 to 0.98) and $n$ values (5 to 25), our algorithm consistently outperforms the baseline. In the Appendix B.4, we further provide the impact of hyperparameter variations on CHAIR$_I$ and F1, along with a more detailed analysis of the results. Moreover, to ensure the completeness of our study, we also examine whether restricting the D-LEAF algorithm to specific layers yields additional gains. The results in Appendix B.4 and D.4 demonstrate that, unlike PAI Liu et al. (2024b) and Deco Wang et al. (2024), D-LEAF consistently achieves significant suppression of hallucinations regardless of whether the layer prior is applied.

### 6 CONCLUSION

We proposed D-LEAF to suppress the hallucinations generated by MLLMs. We propose a two-stage localization and correction algorithm: first, we use the Layer Image Attention Entropy to identify anomalous modules during the forward pass; then, we apply the Image Attention Focus to rank that layer's heads and selectively correct the lowest-performing ones. Our experiments demonstrated that D-LEAF outperforms existing methods in reducing hallucinations across various MLLMs. This work highlights the potential of attention modules to enhance the output reliability of MLLMs and provides mechanistic insights into their operation.

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

# A    STRUCTURE OF THE APPENDIX

The appendix is structured as follows:

Appendix B details the datasets used in our validation experiments as well as the baseline methods for comparison. It also provides additional experiments that complement the main text, including throughput estimates across different models, ablation studies, and related analyses.

Appendix C details the statistical tools employed in this work, along with additional experimental instruments such as the Logit Lens.

Appendix D presents the overall framework of D-LEAF, demonstrating both the soundness and motivation of our proposed metrics, and further verifying their generalizability across model architectures. It also includes supplementary analyses on the role of layer priors in the algorithm and introduces the concept of visual processing layers.

Appendix E provides qualitative case studies showcasing high-quality answers generated by models after applying our algorithm.

Appendix F describes the usage of Large Language Models in the paper.

# B    IMPLEMENTATION DETAILS

## B.1    DATASET

**CHAIR.** The Caption Hallucination Assessment with Image Relevance (CHAIR) metric provides per-image ground-truth object annotations for image captioning, flagging any model-generated object not in the reference set as a hallucination.

**POPE.** The Polling-based Object Probing Evaluation (POPE) evaluates hallucinations in visual question answering by querying "Is there a <object> in the image?" using three object-sampling strategies:

- Random: uniformly drawn from the full dataset.
- Popular: selected from the most frequent objects.
- Adversarial: chosen for strong semantic relevance to the image.

**MMHal-Benchmark.** MMHal-Bench comprises 96 image–question pairs spanning 12 COCO-derived object meta-categories and eight question types (attributes, adversarial, comparison, counting, spatial relations, environment, holistic descriptions, and others), providing a rigorous testbed for evaluating model hallucination in challenging examples.

In addition, to evaluate the effectiveness of our method, we tested the number of tokens output per second by the model in each of the three models.

## B.2    BASELINES

**Greedy Search and Nuclear Sampling.** Traditional decoding strategies that are widely used in sequence generation tasks.

**SPIN and PAI.** SPIN (Sarkar et al., 2025) and PAI (Liu et al., 2024b): The latest SOTA approach leverages attention-head mechanisms to effectively suppress hallucinations.

**VCD.** VCD (Leng et al., 2024): A technique that introduces noise into images to create amateur models for contrastive decoding.

**DAMRO.** DAMRO (Gong et al., 2024): This method leverages the ViT's CLS token to selectively filter out high-attention background outliers and eliminate their influence during decoding.

We used the parameters provided in the open source version of these methods.

## B.3 Throughput Estimation

In the main text, we demonstrated that on LLaVA our algorithm achieves throughput closest to greedy search among all methods. To further validate its effectiveness, we measured throughput on MiniGPT-4 and Shikra. As shown in Figure 10, our approach still incurs the smallest throughput degradation while maintaining high hallucination suppression rates and preserving output detail.

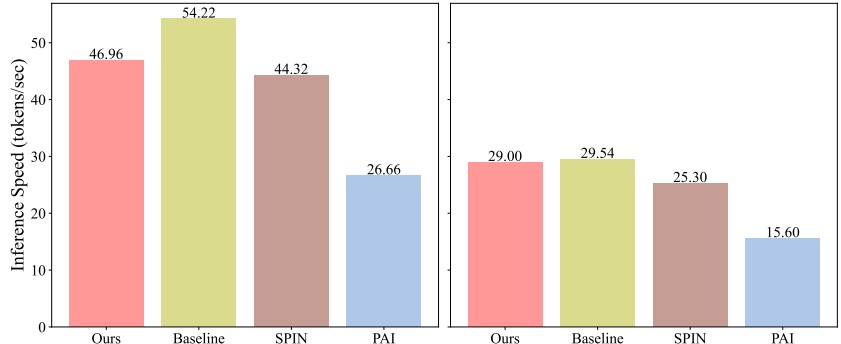

Figure 10: Throughput comparison with existing methods given by the number of tokens generated per second in MiniGPT-4 (left) and Shikra (right).

## B.4 Ablation Study

D-LEAF is a two-stage localization–correction algorithm for hallucination suppression. In the first stage, localization is performed by comparing the Layer Image Attention Entropy (LIAE) with the current Best Attention Score (BAS) to identify abnormal layers. This process does not require any additional hyperparameters, though layer priors, $L$, can be applied during this phase, which restricts localization to specific layers and improve detection accuracy. In the second stage, correction is carried out by ranking the attention heads within the identified abnormal layers using the Image Attention Focus (IAF). The lowest-performing $n$ heads are then selected and refined through a mixing adjustment with a correction coefficient $\gamma$.

Table 3 shows that incorporating Layer Priors, $L$, into our algorithm yields a 10% improvement in hallucination reduction compared to the variant without layer priors. Regardless of whether layer priors are applied, our method consistently achieves the best performance among hallucination suppression algorithms. However, we observe that while hallucinations decrease, the model's F1 score also drops—an issue similarly reported in other attention-head-based suppression methods (Liu et al., 2024b; Sarkar et al., 2025). In Appendix D.4, we further evaluate this phenomenon and find that introducing layer priors can mitigate this decline, enabling hallucination reduction while maintaining or even improving F1. Nonetheless, a fundamental trade-off remains between the two.

Table 3: Ablation Study of Layer Prior. The best result is highlighted in bold, and the second-best is underlined.

| Model | $L$ | $C_S$ | $C_I$ | F1 |
|---|---|---|---|---|
| LLAVA | baseline | 47.08 ±1.54 | 13.00 ±0.59 | **77.22** ±0.66 |
| | with L | **23.44** ±2.63 | **6.72** ±0.49 | 74.88 ±1.51 |
| | without L | 26.20 ±2.63 | 8.30 ±1.32 | 74.28 ±0.81 |
| MiniGPT-4 | baseline | 34.00 ±1.98 | 10.82 ±0.59 | **69.60** ±0.99 |
| | with L | **11.56** ±1.69 | **4.72** ±0.95 | 66.06 ±0.99 |
| | without L | 14.60 ±1.51 | 6.76 ±1.13 | 66.52 ±0.61 |
| Instrutblip | baseline | 48.12 ±2.98 | 14.18 ±1.12 | **73.92** ±0.82 |
| | with L | 22.28 ±1.32 | **7.82** ±2.88 | 70.06 ±0.80 |
| | without L | **22.44** ±2.75 | 8.48 ±5.93 | 70.74 ±1.13 |

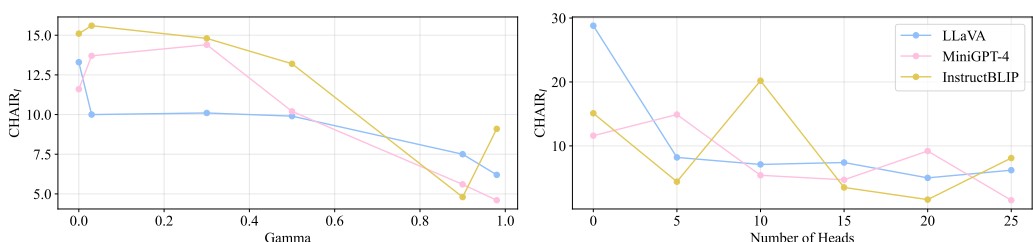

Figure 11: Ablation Study results with $\text{CHAIR}_I$ for hyperparamter $\gamma$ and $n$.

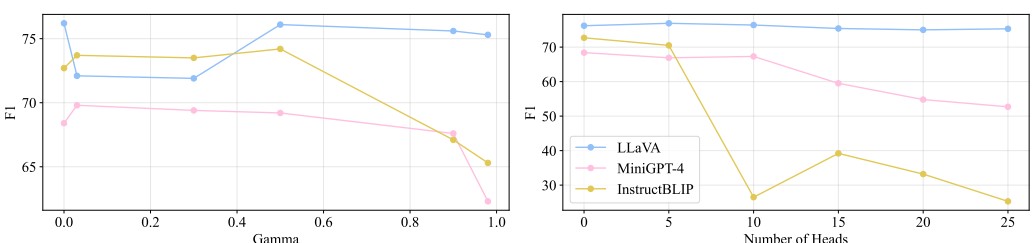

Figure 12: Ablation Study results with F1 for hyperparamter $\gamma$ and $n$.

In addition to the analysis in the main text on the impact of the mixing coefficient $\gamma$ and the number of suppressed attention heads $n$ on $\text{CHAIR}_S$, we further report here the influence of these hyperparameters on $\text{CHAIR}_I$ and F1, as shown in Figure 11 and 12. The results demonstrate the strong robustness of our method: across a wide range of $\gamma$ values (0.03 to 0.98), our algorithm consistently outperforms the baseline. We further observe that increasing $\gamma$ leads to a significant improvement in hallucination suppression across all three models. However, overly large $\gamma$ values result in a drop in F1 score, whereas appropriately chosen $\gamma$ achieves a favorable balance—reducing hallucinations while maintaining high F1 performance.

As for the impact of the number of corrected attention heads, $n$, on hallucination reduction, we find that in models such as InstructBLIP, which leverage a learnable querying transformer to establish vision–language connections with only 32 image tokens as MLLM input, correcting even a small subset of attention heads achieves strong suppression performance. However, as the number of corrected heads increases, the model's output capability deteriorates significantly: the F1 score drops sharply and the generated responses become markedly shorter. We attribute this to excessive correction disrupting the model's normal output dynamics, causing it to prefer shorter responses as a way of avoiding hallucinated tokens. Interestingly, contrary to this trend, in LLaVA, which employs an MLP to map the vision branch output into 576 image embeddings, the number of corrected attention heads does not substantially affect model performance.

## C  PRELIMINARY

In this section, we provide a detailed overview of the analysis tools employed, such as LogitLens and the Wilcoxon signed-rank test, and the forward process of MLLMs.

### C.1  MODEL FORWARD PROCESS

The language decoder module here consists of multiple transformer components (Vaswani et al., 2017). Each transformer block comprises two sublayers: a multi-head attention (MHA) mechanism and a feed-forward network (FFN). MHA begins by taking the combined text and image embedding vectors $X \in R^{N \times d}$ as input, projecting them into the query $(Q)$, key $(K)$, and value $(V)$ spaces, and then computing the output of the MHA module. The output of MHA is fed into the FFN module, and the final output of the current encoder block is obtained through residual flow, as shown in equation 7 and equation 8.

$$x_n^{mid,l} = \sum_{h=1}^{H} \text{Attn}^{(l,h)}(X_{\leq n}^{l-1})W_U \tag{7}$$

$$f(x) = \sum_{l=1}^{L} x_n^{mid,l} + \sum_{l=1}^{L} \text{FFN}^l(x_n^{mid,l})W_U + x_n W_U \tag{8}$$

Therefore, we are able to suppress hallucinations by making corrections in the MHA module during the forward process without modifying the model architecture or adding additional training.

## C.2 LOGITLENS

LogitLens (nostalgebraist, 2020) is an interpretability technique that directly maps each hidden state $x^l$ to the model's vocabulary distribution by first applying the LayerNorm transformation and then projecting through the unembedding matrix $W_U$, as shown in equation 9.

$$\text{LogitLens}(x^l) = \text{LayerNorm}(x^l)W_U \tag{9}$$

We used LogitLens to analyze the probability curves of the model's target logit under both selective and indiscriminate correction, which enabled us to recognize that indiscriminate correction did not genuinely take effect in certain scenarios.

## C.3 WILCOXON SIGNED-RANK TEST

The Wilcoxon signed-rank test is a non-parametric method for assessing whether the median difference between paired samples is zero. Given paired observations $(x_i, y_i)$, we compute the differences

$$d_i = x_i - y_i,$$

exclude any zero differences, and rank the remaining absolute values $|d_i|$ to obtain ranks $R_i$. The test statistic is then defined as

$$W = \sum_{i=1}^{n} \text{sign}(d_i)\, R_i. \tag{10}$$

Under the null hypothesis that the distributions of $x_i$ and $y_i$ are identical, $W$ has a known sampling distribution, from which we derive a two-sided $p$-value to determine significance.

## C.4 ISOTONIC REGRESSION

Isotonic regression is a non-parametric technique for fitting a monotonic (non-decreasing) function to a set of paired observations $(x_i, y_i)$. It estimates values $f_i$ by solving

$$\min_{f_1 \leq f_2 \leq \cdots \leq f_n} \sum_{i=1}^{n} w_i\,(y_i - f_i)^2, \tag{11}$$

subject to the ordering constraints $f_i \leq f_{i+1}$, where $w_i$ are optional non-negative weights. This problem is efficiently solved using the Pool Adjacent Violators Algorithm (PAVA), which produces a piecewise-constant fit that enforces the desired monotonic relationship.

## C.5 SPEARMAN CORRELATION COEFFICIENT

The Spearman correlation coefficient $\rho$ is a non-parametric measure of rank correlation that evaluates the strength and direction of a monotonic relationship between two variables. Given paired observations $(x_i, y_i)$ for $i = 1, \ldots, n$, we first convert them to ranks $R(x_i)$ and $R(y_i)$, and then compute

$$\rho = \frac{\sum_{i=1}^{n}\left(R(x_i) - \overline{R_x}\right)\left(R(y_i) - \overline{R_y}\right)}{\sqrt{\sum_{i=1}^{n}\left(R(x_i) - \overline{R_x}\right)^2}\,\sqrt{\sum_{i=1}^{n}\left(R(y_i) - \overline{R_y}\right)^2}}, \tag{12}$$

where $\overline{R_x} = \frac{1}{n}\sum_{i=1}^n R(x_i)$ and $\overline{R_y} = \frac{1}{n}\sum_{i=1}^n R(y_i)$. Alternatively, when there are no tied ranks, it can be expressed as

$$\rho = 1 - \frac{6\sum_{i=1}^n d_i^2}{n(n^2-1)}, \quad d_i = R(x_i) - R(y_i).$$

The coefficient ranges from $-1$ (perfect negative correlation) to $+1$ (perfect positive correlation), with $\rho = 0$ indicating no monotonic association.

## D   DETAILS OF D-LEAF FRAMEWORK

In this section, we first provide an an empirical analysis on the metrics in D-LEAF, followed by verification of the generalisability of various metrics in D-LEAF under other model architectures (Shikra) and the overall process of the D-LEAF algorithm.

### D.1   EMPIRICAL ANALYSIS ON ENTROPY AND FOCUS

Previous studies have suggested that insufficient and overly dispersed visual-stream attention is one of the primary causes of hallucination in MLLMs. To validate this claim, we examine Shikra and MiniGPT-4, comparing the degree of attention to image regions and the entropy of attention distributions across different layers when the models generate hallucinated versus factual tokens.

We randomly sampled 500 images from the COCO2014 validation set and, for each image, extracted the model's attention matrices when generating hallucinated versus ground-truth tokens, respectively. We then computed the attention scores over the image region for both cases. The results are shown in Figure 13. We observe that, in both models, across all layers, image-region attention for ground-truth tokens is higher than hallucinated tokens.

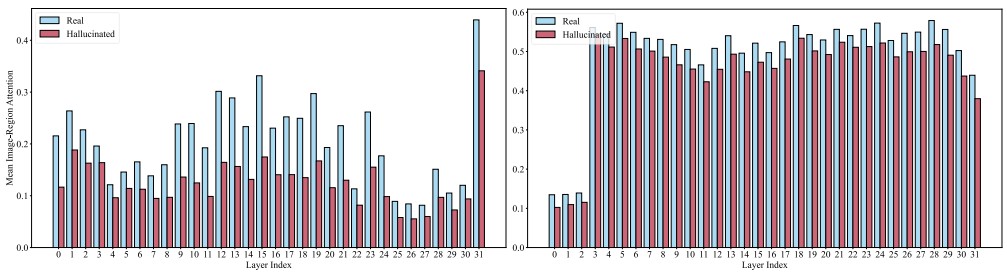

Figure 13: Comparison of attention scores for real words and hallucinated words in the image region in MiniGPT-4 (left) and Shikra (right).

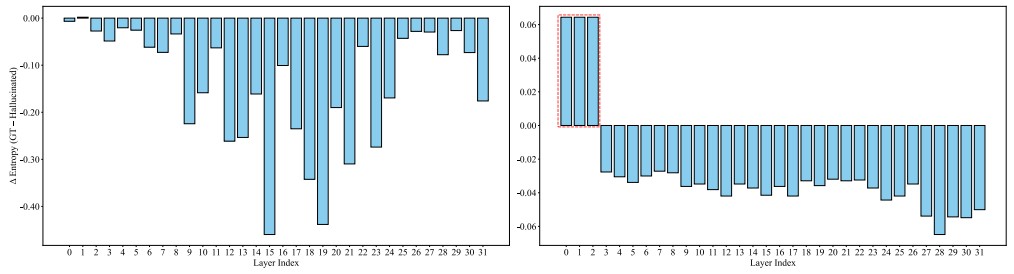

Figure 14: Layer-Wise Entropy Difference: Ground-Truth minus Hallucinated in MiniGPT-4 (left) and Shikra (right).

Similarly, after applying softmax normalization to the attention matrices for ground-truth and hallucinated tokens, we computed the mean image-region entropy across all attention heads at each layer and plotted their per-layer difference as a bar chart in Figure 14. Because the raw entropy values are very close, we scaled the y-axis by a factor of $10^3$. We could observe that, across all

layers, image-region entropy for real tokens is lower than hallucinated tokens. However, in some architectures, like shikra, we observe that in certain layers the average entropy is actually higher for real words than for hallucinated terms. This further confirms that the indiscriminate modification of attention heads, as previously discussed, is suboptimal.

## D.2 METRIC VALIDATION AND COMPARISON

In D-LEAF, we employ the Layer Image Attention Entropy (LIAE) to detect abnormal layers. Once abnormal layers are identified, we rank attention heads using the Image Attention Focus (IAF) to select those requiring correction. In this section, we present four sets of experiments demonstrating that for abnormal layer detection, using LIAE alone outperforms either IAF or a combined metric, whereas for head localization, using IAF alone yields better performance than either IAE or the combined approach.

Inspired by the previous section, we propose layer image attention as a comparison metric for anomaly layer detection.

$$\text{LIAF}^{(l)} = \sum_{n=1}^{N} \text{MAM}_n^{(l)} \tag{13}$$

To evaluate the significance of the metric, which independent and non-normally distributed across real and hallucinated tokens, we apply the Wilcoxon signed-rank test (Wilcoxon, 1992) as in main text. With $p < 0.001$, in Figure 15 we confidently observe in MiniGPT-4 that hallucinated tokens exhibit significantly lower LIAF compared to real tokens.

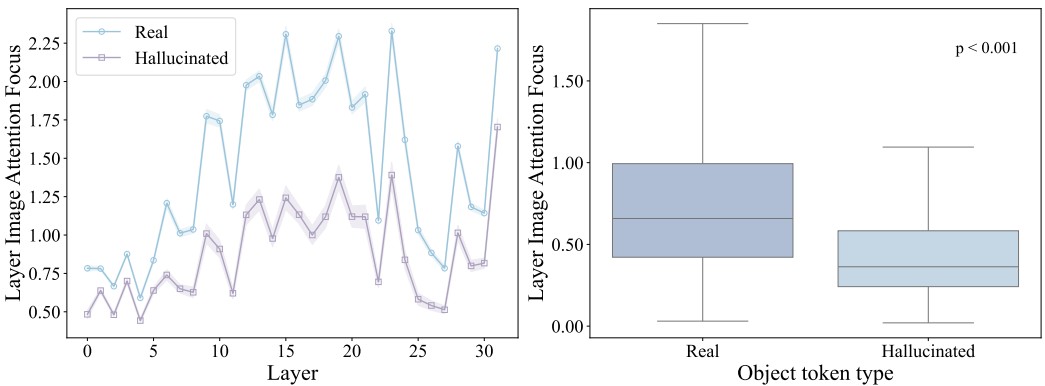

Figure 15: LIAF change curves (left) and distributions , together with the Wilcoxon signed-rank test results (right) for hallucinated versus real words generated by MiniGPT-4.

While we have already confirmed the individual effectiveness of LIAE and LIAF, combining them necessitates conducting a correlation analysis between the two metrics. Since these two metrics do not conform to a normal distribution, we computed the Spearman correlation coefficient between LIAE and LIAF, obtaining $\rho = -0.85$ with $p < 0.001$, which indicates that there is a high negative correlation between LIAE and LIAF. To further characterize their relationship, we fitted an isotonic regression—shown as the solid blue curve in Figure 16 left closely follows, yet remains slightly above, the idealized gray dashed line denoting perfect negative correlation. We obtained the same fitting results for attention head localization metrics: Image Attention Entropy (IAE) and Image Attention Focus (IAF) as illustrated in Figure 16 right.

Based on the above experiments, we can conclude that there is a strong negative correlation between LIAE and LIAF. Therefore, we propose Layer Image Attention Score (LIAS) as a comprehensive indicator for hallucination detection:

$$\text{LIAS}^{(l)} = \alpha \, \text{LIAE}^{(l)} - (1 - \alpha) \, \text{LIAF}^{(l)} \tag{14}$$

To comprehensively evaluate the capability of the three proposed metrics in detecting abnormal layers, we conducted experiments on the CHAIR dataset using LLaVA, MiniGPT-4, and Shikra, as

shown in the table 4. The results demonstrate that selecting LIAE alone as the primary indicator yields the strongest hallucination suppression. However, as LIAF is incorporated, the suppression effect gradually diminishes: once $\alpha$ exceeds 0.5, all three models produce identical detection results. We attribute this to the substantial numerical disparity between LIAE and LIAF, which causes LIAF to increasingly dominate in the mixed metric, whereas LIAE is inherently more sensitive to abnormalities. Therefore, we use LIAE exclusively during detection.

Table 4: Ablation study of detection coefficient $\alpha$. The best result is highlighted in bold, and the second-best is underlined.

| $\alpha$ | LLaVA | | | MiniGPT-4 | | | Shikra | | |
|---|---|---|---|---|---|---|---|---|---|
| | $C_S$ | $C_I$ | F1 | $C_S$ | $C_I$ | F1 | $C_S$ | $C_I$ | F1 |
| 0.0 | **20.6** | **6.2** | 75.3 | **12.6** | **5.4** | 67.3 | **25.2** | **10.2** | 62.5 |
| 0.3 | 33.0 | 12.7 | **76.9** | 35.4 | 10.7 | 69.0 | 35.2 | 12.7 | **67.3** |
| 0.5 | 32.0 | 10.3 | 75.0 | 35.4 | 10.7 | 69.5 | 35.4 | 13.4 | 66.9 |
| 0.7 | 32.0 | 10.3 | 75.0 | 35.4 | 10.7 | 69.5 | 35.0 | 13.2 | **67.3** |
| 1.0 | 32.0 | 10.3 | 75.0 | 35.4 | 10.7 | 69.5 | 35.0 | 13.2 | **67.3** |

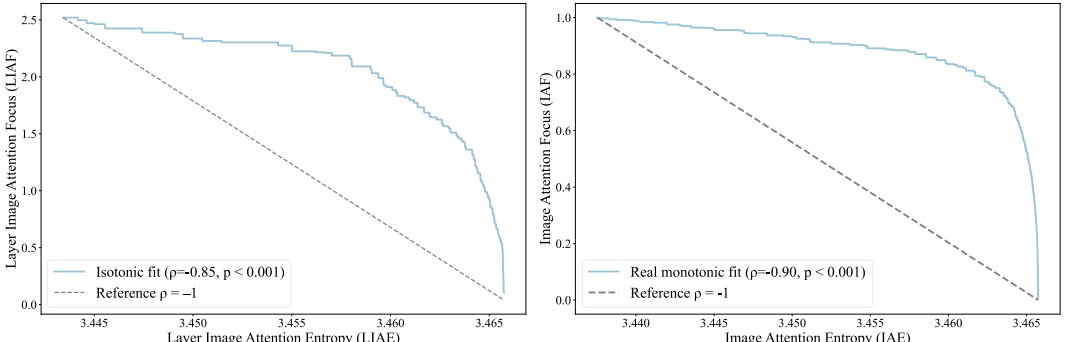

Figure 16: Isotonic Regression fit of LIAE against LIAF (left) and IAE against IAF (right) in MiniGPT-4.

In the correction stage, we additionally introduce two comparative metrics, namely the Image Attention Entropy (IAE) and the Image Attention Score (IAS).

$$\mathrm{IAE}_h^{(l)} = -\sum_{n=1}^{N} \mathrm{P}(A_{h,n}^{(l)})\log \mathrm{P}(A_{h,n}^{(l)}) \tag{15}$$

$$\mathrm{IAS}_h^{(l)} = \beta\,\mathrm{IAF}_h^{(l)} + (1-\beta)\,\mathrm{IAE}_h^{(l)} \tag{16}$$

We visualize the distributional differences of IAE when the model generates hallucinated versus factual tokens, as shown in Figure 17. The results reveal that the discrepancies across attention heads are extremely subtle, appearing only beyond the fifth decimal place. We further repeat the CHAIR experiment and find that once IAE is incorporated, the correction process leads the model to malfunction, as it mistakenly identifies and modifies the wrong attention heads. For these reasons, we use IAF exclusively during correction.

### D.3 EVALUATION OF METRIC GENERALIZABILITY

To further verify the validity and effectiveness of our proposed metrics across different model architectures, we conducted supplementary experiments on other model architecture and analyzed the results. We first plot the distributions of LIAE and IAF in Shikra like the main context in Figure 18.

Additionally, we plotted the LIAF curve and distributions in Figure 19 and performed Wilcoxon signed-rank tests on both metrics for hallucinated versus ground-truth tokens. At p < 0.001, hallucinated tokens exhibit significantly lower LIAF than ground-truth tokens.

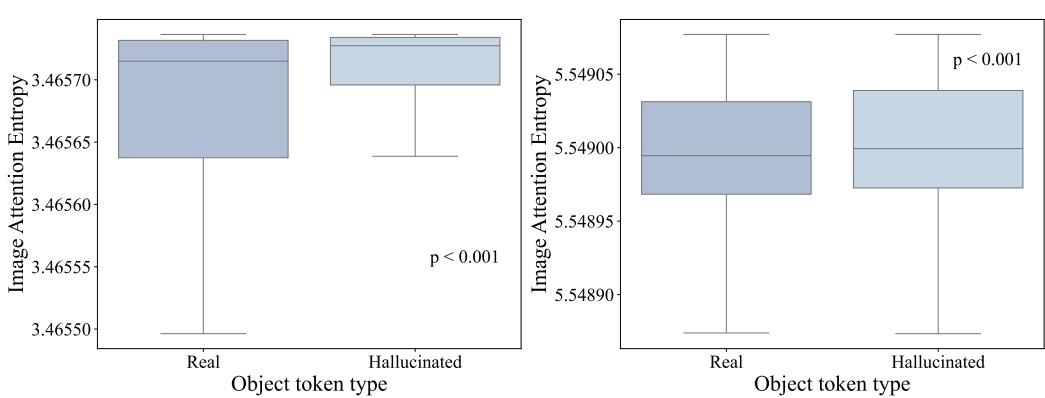

Figure 17: IAE distributions , together with the Wilcoxon signed-rank test results for hallucinated versus real words generated by MiniGPT-4 (left) and Shikra (right).

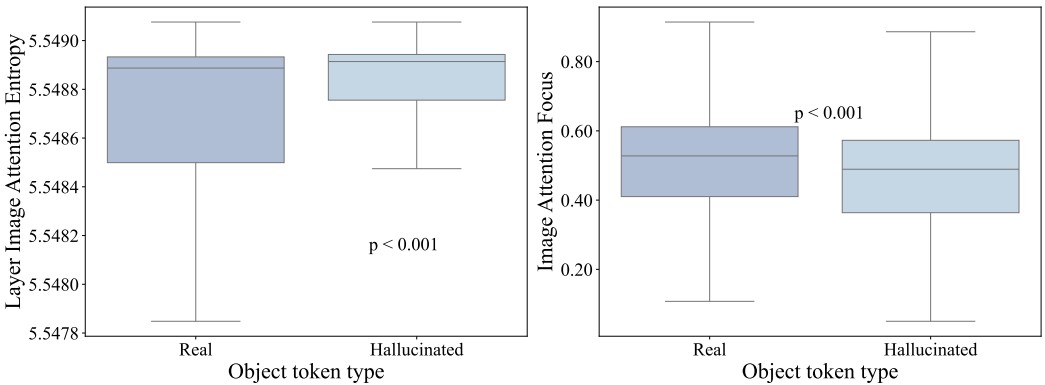

Figure 18: The distributions of LIAE (left) and IAF (right) for hallucinated versus real words, together with the Wilcoxon signed-rank test results in Shikra.

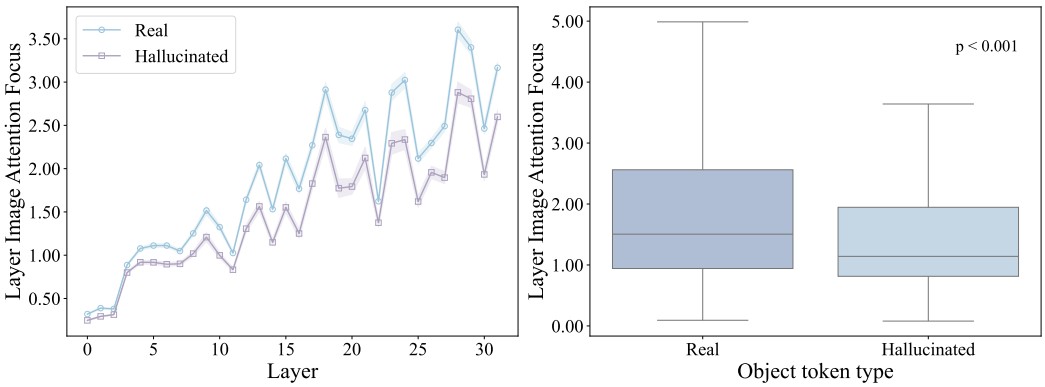

Figure 19: LIAF change curves (left) and distributions , together with the Wilcoxon signed-rank test results (right) for hallucinated versus real words generated by Shikra.

Furthermore, we evaluated the correlation between LIAE and LIAF and fitted it with isotonic regression, as shown in Figure 20 left. The gray dashed line denotes a perfect negative correlation, and the blue curve represents the fitted regression; we observe a correlation of –0.88 at p<0.001.

After confirming that LIAE and LIAF generalize well for identifying anomalous layers, we further evaluated the generalizability of Image Attention Entropy (IAE) and Image Attention Focus (IAF) for pinpointing anomalous attention heads. The distribution and significance test of IAE is presented in Figure 17: for each attention head, Image Attention Entropy (IAE) in ground-truth tokens is significantly lower than in hallucinated tokens (p < 0.001).

Similarly, we applied isotonic regression fitting; as shown in Figure 20 right, although our curve does not perfectly align with the gray dashed line denoting a perfect negative correlation, it still exhibits a clear negative trend, with a correlation coefficient of –0.68 (p<0.001).

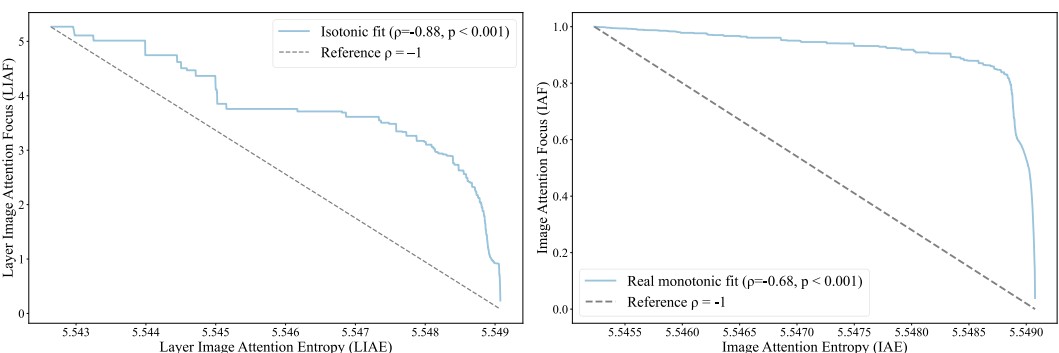

Figure 20: Isotonic Regression fit of LIAE against LIAF (left) and IAE against IAF (right) in Shikra.

In summary, on Shikra we observe the consistent patterns as in MiniGPT-4: ground-truth tokens exhibit significantly lower LIAE and IAE and significantly higher LIAF and IAF compared to hallucinated tokens. Furthermore, both the LIAE–LIAF and IAE–IAF pairs show strong negative correlations, supporting their joint use as detection metrics for anomalous layers and heads.

### D.4 VISUAL PROCESSING LAYER

In the Appendix B.4, we conducted an ablation study on the Layer Prior and found that even without incorporating it, our algorithm already achieves state-of-the-art performance. However, introducing the layer prior further reduces hallucination rates. In this section, we provide a more fine-grained analysis of the Layer Prior and introduce the concept of visual processing layers.

We first evaluated the effect of varying the number of corrected layers on three models: LLaVA, MiniGPT-4, and Shikra, measuring changes in hallucination rate and F1 score, as shown in Table 5. The results demonstrate that regardless of the partitioning strategy, our method consistently achieves a substantial reduction in hallucination rates. Moreover, by tuning the number of corrected layers, our approach is able to maintain high F1 scores while suppressing hallucinations, further confirming both the robustness and strong transferability of our algorithm.

In addition, our experiments reveal that applying corrections within layers 0–25 often yields the most substantial reduction in hallucinations, albeit at the cost of some F1 degradation. We hypothesize that this effect arises because the algorithm is restricted to the visual processing stages: while the model becomes highly effective at distinguishing objects during visual processing, in the subsequent language generation phase it tends to produce shorter outputs to avoid hallucinated tokens, thereby leading to a decline in F1 score. To confirm this, we ran a simple experiment to confirm that all three architectures integrate visual features primarily in layers 0–25.

We tracked the trajectories of the top 90% percentile logits across all layers of MiniGPT-4 and Shikra (Figure 21 and 22). The curves plateau around layer 26, suggesting that content integration and reasoning are effectively completed by the end of the first 25 layers. Accordingly, our hallucination

detection and correction mechanisms are concentrated on these initial layers, which confirm the prior hypothesis.

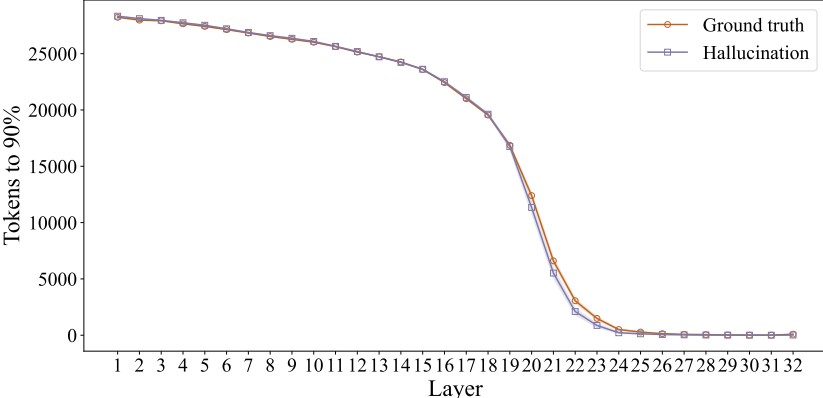

Figure 21: The trajectories of the top 90% percentile logits across all layers of MiniGPT-4.

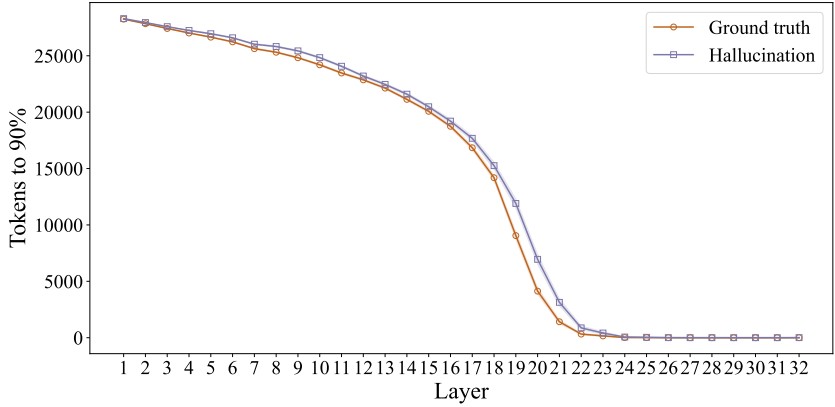

Figure 22: The trajectories of the top 90% percentile logits across all layers of Shikra.

Table 5: Ablation study of correction layers across different models. The best result is highlighted in bold, and the second-best is underlined.

| Layer | LLaVA | | | MiniGPT-4 | | | Shikra | | |
|-------|-------|-------|------|-----------|------|------|--------|------|------|
| | $C_S$ | $C_I$ | F1 | $C_S$ | $C_I$ | F1 | $C_S$ | $C_I$ | F1 |
| 0–10 | 24.0 | 7.0 | 75.1 | 18.8 | **5.6** | 67.6 | 47.2 | 15.2 | 70.3 |
| 5–25 | 37.0 | 11.2 | **77.7** | **12.6** | **5.6** | 67.3 | 53.2 | 14.8 | **74.8** |
| 0–25 | **20.6** | **6.2** | 75.3 | 21.8 | 6.0 | 68.1 | **25.2** | **10.9** | 62.5 |
| 26–31 | 37.0 | 11.2 | 74.7 | 33.0 | 10.6 | **71.7** | 58.0 | 16.3 | 73.4 |
| 0–31 | 22.2 | 7.8 | 72.5 | 16.6 | 6.9 | 65.6 | 28.2 | 11.0 | 64.1 |

## D.5 FULL PROCEDURE OF D-LEAF

Algorithm 1 and Figure 23 shows the full procedure of D-LEAF.

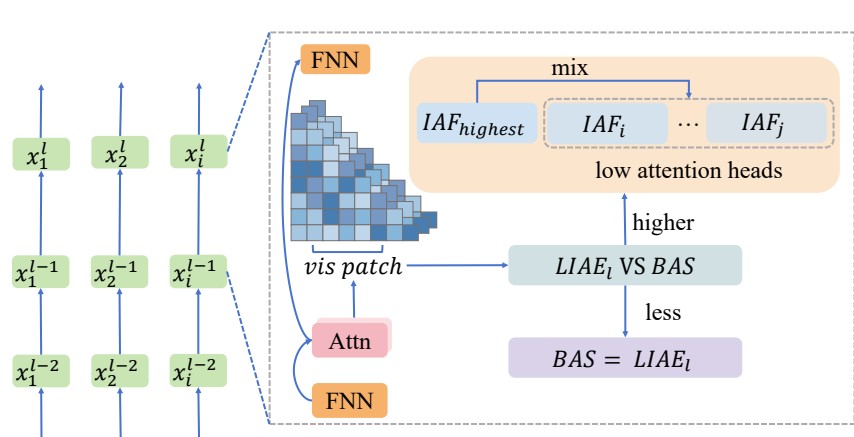

Figure 23: Architecture of our D-LEAF.

---

**Algorithm 1** Dynamic Layer-wise Entropy and Attention Fusion

---

**Input**: N input tokens, consisting of both text and vision tokens, each of embedding dimension d, $X \in \mathrm{R}^{(N \times d)}$

**Parameter**: Optional list of parameters

**Output**: The probability distribution of the next token.

1: Let $l = 0$, $\mathrm{BAS}_{best} = \mathrm{Inf}$, $A_{tot} = \mathrm{None}$.
2: **while** $l < L$ **do**
3:     $Q, K, V = W_Q X, W_K X, W_V X$
4:     **for** $h \in H$ **do**
5:         $A_h = \frac{Q^h (K^h)^T}{\sqrt{d_k / H}}$
6:         **if** $A_{tot} = \mathrm{None}$ **then**
7:             $A_{tot} = A_h$
8:         **else**
9:             $A_{tot} = \mathrm{concat}(A_{tot}, A_h)$
10:         **end if**
11:     **end for**
12:     **if** $l \in (l_s, l_e)$ **then**
13:         $\mathrm{BAS}_l = \mathrm{LIAE}^l$
14:         **if** $\mathrm{BAS}_{best} < \mathrm{BAS}_l$ **then**
15:             $\mathrm{BAS}_{best} = \mathrm{BAS}_l$
16:         **else**
17:             $A_{set} = \mathrm{sort}(A_{tot}, \mathrm{key} = \mathrm{IAF}, \mathrm{asc} = \mathrm{True})$
18:             **for** $A_h \in A_{set}[0 : n]$ **do**
19:                 $A_{h,v} = A_h[I_{start} : I_{end}]$
20:                 $A_{best,v} = A_{set}[-1][I_{start} : I_{end}]$
21:                 $A_{h,v} = \gamma A_{best,v} + (1 - \gamma) A_{h,v}$
22:             **end for**
23:         **end if**
24:     **end if**
25:     $X = \mathrm{FFN}(A_{tot} V)$
26:     $l = l + 1$
27: **end while**
28: **return** $X$

---

# E   CASE STUDY

Additional case studies across diverse MLLMs are provided to demonstrate the effectiveness of D-LEAF. Specifically, Figures 24, 25 and 26 respectively present further instances of hallucination corrections by our method for LLaVA-1.5, MiniGPT-4 and Shikra.

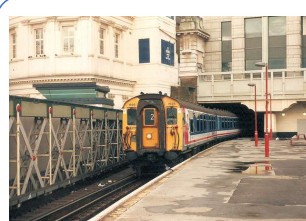

**Input Image**

**Instruction:** Please describe the image in detail.

**LLaVA-1.5 (Greedy):**
The image features a yellow and red train parked at a train station. The train is positioned on the tracks, occupying a significant portion of the scene. The train station appears to be a busy place, with several people scattered around the area. Some of them are standing close to the train, while others are further away. In addition to the train and people, there are a few objects in the scene. A bench can be seen near the right side of the image, and a handbag is placed on the ground close to the center. A clock is also visible in the background, providing information about the time for the passengers.

**LLaVA-1.5 (D-LEAF):**
The image shows a yellow-fronted passenger train arriving at a station platform. The train has a blue and red stripe along its side and is positioned on curved tracks next to the platform. The surrounding area includes station buildings and a covered section where the train is emerging from. The platform appears mostly empty, with no visible passengers nearby.

---

**Instruction:** Please describe the image in detail.

**LLaVA-1.5 (Greedy):**
The image features a man and a woman standing together, both holding wine glasses in their hands. They are posing for a picture, likely celebrating a special occasion or simply enjoying each other's company. The man is wearing glasses, and the woman is wearing a scarf. In the background, there are two dining tables, one on the left side and another on the right side of the scene. A clock can be seen hanging on the wall, and a chair is placed near the left side of the room. Additionally, there are two bottles in the scene, one located near the center and the other on the right side.

**LLaVA-1.5 (D-LEAF):**
The image features a man and a woman standing next to each other, both holding wine glasses filled with champagne. They are posing for a picture, and the woman is wearing a black shirt. The man is wearing a black shirt and a scarf around his neck. The couple appears to be enjoying a celebration or a special occasion.

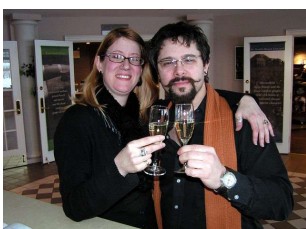

**Input Image**

---

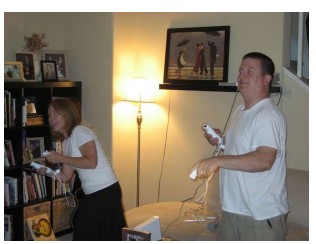

**Input Image**

**Instruction:** Please describe the image in detail.

**LLaVA-1.5 (Greedy):**
The image features a man and a woman standing in a living room, both holding Wii game controllers and playing a video game together. They are standing close to each other, with the woman on the left and the man on the right. The living room is furnished with a couch in the background and a dining table nearby. There are several books scattered around the room, with some placed on the dining table and others on the floor. A clock is also visible on the wall, adding to the cozy atmosphere of the living space.

**LLaVA-1.5 (D-LEAF):**
The image shows two people in a living room playing a video game. They are holding motion controllers and appear to be enjoying the activity, with smiles on their faces. The room has a bookshelf, a standing lamp, and a framed painting on the wall, creating a casual and relaxed atmosphere.

Figure 24: D-LEAF's performance on reducing hallucinations of LLaVA-1.5.

# F   LLM USAGE

We used large language models (LLMs) as a general-purpose tool only for minor editing and language polishing. LLMs were not involved in research ideation, data analysis, or the generation of scientific insights. All intellectual contributions, including the conception, design, and interpretation of the work, are solely those of the authors.

1296
1297
1298
1299
1300
1301
1302
1303
1304
1305
1306
1307
1308
1309
1310
1311
1312
1313
1314
1315
1316
1317
1318
1319
1320
1321
1322
1323
1324
1325
1326
1327
1328
1329
1330
1331
1332
1333
1334
1335
1336
1337
1338
1339
1340
1341
1342
1343
1344
1345
1346
1347
1348
1349

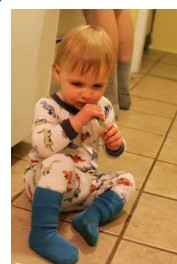

**Input Image**

**Instruction:** Please describe the image in detail.

**MiniGPT-4 (Greedy):**
The image shows a small child sitting on the floor in a kitchen, wearing a pair of blue socks and a white and blue striped shirt. The child is holding a toothbrush in one hand and brushing their teeth with the other. The child's parents are standing behind them, watching and helping them brush their teeth. The floor is tiled and there is a sink and faucet in the background.

**MinGPT-4 (D-LEAF):**
The image shows a young child sitting on a tiled floor, wearing patterned pajamas with colorful designs and blue socks. The child appears to be holding and examining a small object, while part of an adult's legs can be seen in the background.

**Instruction:** Please describe the image in detail.

**MiniGPT-4 (Greedy):**
The image shows a large, white and blue airplane flying in the sky against a clear blue background. The plane has a yellow tail and two white engines on the back. The plane is flying at a high altitude and appears to be in the process of taking off. The sky is clear and there are no clouds in the sky. The plane is flying in a straight line and appears to be heading towards the horizon.

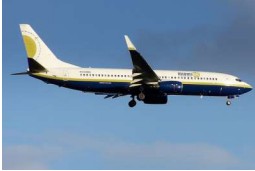

**Input Image**

**MiniGPT-4 (D-LEAF):**
The image shows a commercial airplane in flight against a clear sky. The plane is painted in white, blue, and yellow colors, with the landing gear extended as if preparing to land. The airline's name is visible on the side of the aircraft.

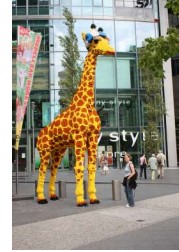

**Instruction:** Please describe the image in detail.

**MiniGPT-4 (Greedy):**
The image shows a large giraffe made out of lego blocks standing in front of a tall building. The giraffe has a long neck and legs, and its body is covered in brown and yellow lego blocks. There is a woman standing next to the giraffe, looking at it with a smile on her face. The woman is wearing a red shirt and black pants, and has short brown hair. The building behind the giraffe has large windows and a modern design. There are several people walking on the sidewalk in front of the building, and a few cars parked on the street.

**MiniGPT-4 (D-LEAF):**
The image shows a large giraffe sculpture built from yellow and orange blocks, standing in an outdoor urban area. The sculpture is positioned near a modern glass building, and several people are walking or standing nearby. A woman is standing close to the sculpture, looking toward the camera, while the surroundings suggest a busy public space.

Figure 25: D-LEAF's performance on reducing hallucinations of MiniGPT4.

**Instruction:** Please describe the image in detail.

**Shikra (Greedy):**
The image features a long blue and yellow train traveling down the train tracks. The train is followed by several cars, including some that appear to be blue and yellow as well, giving the scene a colorful and lively atmosphere. There are multiple people near the train tracks, possibly observing or waiting for the train to pass. Some of them are standing closer to the front of the train, while others are positioned further back. The overall scene conveys a busy and active transportation hub.

**Shikra (D-LEAF):**
The image features a train traveling down the railroad tracks. The train is quite long, occupying a significant portion of the view. There are multiple train tracks visible in the scene, indicating a busy transportation hub.

---

**Instruction:** Please describe the image in detail.

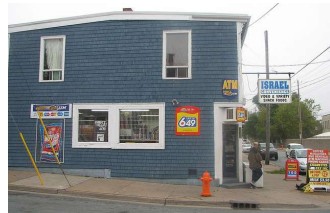

**Shikra (Greedy):**
The image features a small blue and gray storefront of a convenience store located on a street corner. The store has a fire hydrant in front of it, adding a touch of safety to the scene. There is a man standing outside the store, possibly a customer or an employee. In the vicinity of the store, several cars are parked on the street, with one car on the left side of the image, another closer to the center, and two more cars on the right side. Additionally, there is a truck parked further back on the right side of the scene.

**Shikra (D-LEAF):**
The image features a man standing in front of a small, blue brick store, which appears to be a convenience store. The store is located on a street corner and has a fire hydrant nearby. There are two ATMs in front of the store, which are likely for customers to withdraw money. In the vicinity, there are multiple vehicles parked on the street. The scene captures the everyday life of a busy urban area with various elements of street life and businesses coexisting.

---

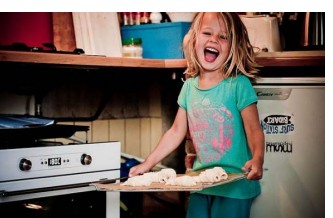

**Instruction:** Please describe the image in detail.

**Shikra (Greedy):**
The image features a little girl standing in a kitchen, holding a tray with several doughnuts on it. She appears to be enjoying her time, possibly preparing something fun for breakfast or dessert. The doughnuts are in various stages of preparation, with some being rolled and others placed on the tray. There are two refrigerators visible in the background, one situated on the left side and the other on the right side of the room. A bottle can be seen on the countertop near the refrigerators, along with an oven placed in the lower left corner of the kitchen.

**Shikra (D-LEAF):**
The image shows a young girl standing in a kitchen, holding a baking tray with unbaked pastries. She is smiling widely and appears very excited. An oven with the door open is in front of her, and various kitchen items can be seen in the background.

Figure 26: D-LEAF's performance on reducing hallucinations of Shikra.

