# OpenReview forum: "D-LEAF: Localizing and Correcting Hallucinations in Multimodal LLMs via Layer-to-head Attention Diagnostics"
_ICLR.cc/2026/Conference — ICLR 2026 Conference Withdrawn Submission_

### Official Review · Reviewer_Yt3F · 2025-10-20

**Soundness:** 3
**Presentation:** 3
**Contribution:** 2
**Rating:** 2
**Confidence:** 5

**Summary:**

This paper proposes a novel inference-time intervention method, D-LEAF. Its core idea is to first localize and then correct. Using two lightweight diagnostic metrics—Layer Image Attention Entropy (LIAE) and Image Attention Focus (IAF)—it dynamically identifies abnormal layers and attention heads that cause hallucinations and selectively corrects only these components.

**Strengths:**

With a clear analysis and strong motivation, the paper explicitly points out that current attention-based methods for hallucination suppression (such as SPIN and PAI) suffer from the problem of "indiscriminate suppression," which can inadvertently harm properly functioning attention heads. Empirical analysis, such as Figures 2 and 3, strongly reveals the uneven distribution of abnormal attention heads across layers, providing a solid basis for the "localize first, then correct" strategy.

**Weaknesses:**

1.The logical coherence of the writing needs improvement. For example, what is the motivation behind the proposed IAF in the abstract? How are ranked heads related to hallucination?

2.The authors claim that previous attention-based hallucination mitigation methods are limited by applying uniform adjustments across layers and heads. However, to my knowledge, there are several related works addressing this issue, such as [MemVR, ICML], [EAH, EMNLP], [see what you told, ICLR], [Cracking the Code of Hallucination in LVLMs with Vision-aware Head Divergence, ACL], and [Devils, CVPR]. I did not see any discussion or citation of these works in the paper.

3.The experimental design is insufficient. The correlation between LIAE, IAF, and hallucination lacks empirical validation.

4.The experiment in Figure 2 on Indiscriminate Modification is insufficient. Additional comparative results with other attention-based hallucination mitigation methods are needed.

5.The statement in Section 3 that “prior attention-based hallucination mitigation methods fail to produce correct answers in semantically ambiguous scenarios” is rather subjective and lacks experimental support. Moreover, the definition of a semantically ambiguous scenario is unclear.

6.The clarity of the methodology section needs improvement: (1) Chapter 4 contains many redundant descriptions; (2) the introduction of MAM and LIAE is unnatural and confusing; (3) the section lacks rigorous mathematical derivation, making the method appear more empirical; (4) IAF seems to directly follow [PAI, ECCV]. In addition, MAM, LIAE, and IAF all appear to lack sufficient theoretical and mathematical depth.

7. The most critical weakness is that this paper does not report the recall of CHAIR and POPE. I guess this work has the same weakness as [PAI, Devils], which is that the recall drops very sharply, which means that the diversity of the model output decreases. This work of directly enhancing image attention through coefficients is unreasonable.

**Questions:**

The overall workload of the paper is insufficient, particularly in the experimental section. The authors are encouraged to include: (1) generalization experiments on more recent baseline LVLMs (e.g., QwenVL2.5/3, VILA, InternVL, and chatgpt-oss); (2) generalization experiments across different model sizes (e.g., 13B, 32B); (3) additional interpretability analyses (e.g., attention map visualization, heatmaps); (4) evaluations on general benchmarks (e.g., GQA, TextVQA, MMStar, MMMU, SEED, ChartQA, etc.); (5) a more thorough efficiency analysis (latency, time cost, memory cost); (6) results on more hallucination benchmarks (e.g., MME); (7) performance comparison with more recent hallucination mitigation methods (e.g., [MemVR, ICML], [CCA, NeurIPS], [EAH, EMNLP], etc.); and (8) a more comprehensive reporting of results, including Recall scores on CHAIR and POPE.

---

### Official Review · Reviewer_CL3U · 2025-10-28

**Soundness:** 3
**Presentation:** 3
**Contribution:** 3
**Rating:** 6
**Confidence:** 3

**Summary:**

This paper presents D-LEAF, a lightweight and task-agnostic framework for mitigating hallucinations in Multimodal Large Language Models (MLLMs).
By introducing two attention-based diagnostics—Layer Image Attention Entropy (LIAE) and Image Attention Focus (IAF)—the authors dynamically localize abnormal layers and heads during inference and apply selective correction with minimal overhead.
The method achieves substantial improvements on multiple benchmarks (up to 53% hallucination reduction) while preserving inference efficiency.
Overall, the paper is clearly written, empirically solid, and proposes an intuitive yet effective way to address a long-standing issue in MLLMs.

**Strengths:**

- The paper introduces a simple yet effective approach to hallucination mitigation without requiring retraining, achieving strong improvements across diverse MLLMs.

- The proposed LIAE/IAF diagnostics provide interpretable insight into where hallucinations arise in the attention hierarchy, bridging a key gap between empirical mitigation and mechanistic understanding.

- The method maintains near-real-time inference speed, making it practical for deployment and comparison-ready with existing attention-based correction methods.

**Weaknesses:**

- The paper does not clearly explain why partial interpolation (rather than full replacement) is theoretically or empirically preferable in Eq. (4).


- It would strengthen the paper to analyze potential side-effects of D-LEAF, such as whether repeated fusion might degrade fine-grained spatial grounding or alter non-hallucinated details over multiple turns.

- The paper could benefit from a clearer ablation or visualization explaining how much each diagnostic (LIAE vs IAF) individually contributes to the final performance.

- In the Related Works section (“Mitigation in MLLMs”), I recommend authors to cite “Do You Keep an Eye on What I Ask? Mitigating Multimodal Hallucination via Attention-Guided Ensemble Decoding” (ICLR 2025), which closely relates to the attention-guided correction idea.

**Questions:**

- In the Mixed Attention Matrix Correction, when γ approaches 1, the corrected head’s attention matrix effectively becomes identical to that of the highest-scoring head. Given that Figure 9 shows performance improving as γ increases, is γ scaling actually necessary in Eq. (4)?

- Is there empirical evidence that high LIAE layers always contain low-IAF heads, or could these metrics occasionally diverge?

- Is there any experimental evidence demonstrating the effectiveness of the Best Attention Score (BAS) mechanism?
It seems rather heuristic, and I’m not fully convinced why this specific update rule is necessary or preferable over alternative thresholding strategies.

---

### Official Review · Reviewer_FDRF · 2025-10-31

**Soundness:** 3
**Presentation:** 3
**Contribution:** 2
**Rating:** 4
**Confidence:** 4

**Summary:**

The paper introduces D-LEAF (Dynamic Layer-wise Entropy and Attention Fusion), a lightweight, training-free framework for localizing and correcting hallucinations in MLLMs

D-LEAF dynamically fuses attention from well-performing heads to correct these problematic ones during inference. The method avoids retraining or extra models, providing a plug-and-play correction mechanism.

**Strengths:**

Lightweight & generalizable: Plug-and-play inference correction applicable to different MLLMs.

Interpretability: Provides convincing explanations for the internal attention failures leading to hallucination.

**Weaknesses:**

Evaluation diversity: Only CHAIR, POPE, and MMHal-Bench are used. More recent benchmarks (e.g., HallusionBench 2025, HAL-Bench RLHF-V) could strengthen generality claims.

Lack of novelty. The attention mechanism is widely known to be correlated with hallucination. Modifying attention weights is also normal an engineer.  A more elegant way is needed for an ICLR paper

Theoretical analysis: LIAE/IAF metrics are empirically defined; a stronger theoretical justification (e.g., linking entropy thresholds to causal influence) would improve rigor.

**Questions:**

Please see weakness paert.

---

### Official Review · Reviewer_CZhE · 2025-11-01

**Soundness:** 3
**Presentation:** 3
**Contribution:** 2
**Rating:** 4
**Confidence:** 3

**Summary:**

This paper studies the problem of hallucination in multimodal large language models (MLLMs), where the generated text conflicts with the visual input.
Previous work has typically attributed this issue to insufficient visual attention. However, existing attention-based detectors and mitigation methods often apply uniform adjustments across all layers and attention heads, thereby obscuring the true sources of error.
The paper first demonstrates that these conventional approaches fail to accurately identify problematic layers. To address this, the authors introduce two diagnostic metrics:
Layer Image Attention Entropy (LIAE), which identifies abnormal layers, and Image Attention Focus (IAF), which ranks attention heads within these layers.
Their analysis shows that LIAE effectively pinpoints faulty layers, while IAF reliably ranks the attention heads that require correction.
Building on these insights, the paper proposes Dynamic Layer-wise Entropy and Attention Fusion (D-LEAF), a task-agnostic, attention-guided inference-time method that dynamically detects and corrects attention errors with minimal computational overhead.

**Strengths:**

1. Experimental results show that D-LEAF achieves a relative improvement of 53% on standard captioning benchmarks, and about 4% gains in accuracy and F1 score on VQA benchmarks.

2. The paper is well written, with clear organization and accessible presentation.

**Weaknesses:**

1. Regarding the case presented in Figure 2, the reviewer understands that it is intended to demonstrate the source of visual hallucination in certain models. The problem is that, for today’s mainstream LVLMs, such simple mistakes almost never occur.
In other words, is the analysis of the hallucination problem in this paper built on an issue that no longer exists in most of today’s advanced open-source models? Or can the authors provide some more complex and realistic cases?

2. Similarly, the experimental models used in the paper—such as Qwen-VL, LLaVA, MiniGPT, and InstructBLIP—are relatively old, most of them published in 2023. Can the authors provide more experimental results on newer visual models, whether they are domain-specific or general-purpose?

**Questions:**

See the content mentioned in the Weaknesses section.

---

### Note · Authors · 2025-11-13

I have read and agree with the venue's withdrawal policy on behalf of myself and my co-authors.